# Protocol for a mixed studies systematic review on the implementation of the recovery approach in adult mental health services

Myra Piat,[1,2,3] Eleni Sofouli,[3] Judith Sabetti,[2,3] Angella Lambrou,[4] Howard Chodos,[5] Catherine Briand,[6] Brigitte Vachon,[6] Janet Curran[7]

For numbered affiliations see end of article.

**Correspondence to**
Dr Myra Piat;
myra.piat@douglas.mcgill.ca

## ABSTRACT

**Introduction** Recovery is integral to mental health planning in G-8 countries including Canada. A recovery-oriented approach to care aims to promote personal empowerment, illness self-management and a life beyond services for people with serious mental illness (SMI), while reducing the financial burden associated with mental illness. Although there is a growing body of literature on recovery, no synthesis of research on the implementation of recovery into mental health services exists.

**Objectives** The objective is to conduct a mixed studies systematic review on the operationalisation of recovery into mental health services for adults with SMI. It will inform the transformation of Canadian services to a recovery orientation, but may be applicable to other countries.

**Methods and analysis** Seven databases including PubMed, Ovid Medline, Ovid Embase, Ovid PsycInfo, CINAHL, the Cochrane Library and Scopus will be searched for peer-reviewed empirical studies published from 1998 to December 2016. Systematic reviews and studies using quantitative, qualitative and mixed methodologies will be included. Secondary searches will be conducted in reference lists of all selected full text articles. Handsearches will also be performed in the tables of contents of three recovery-focused journals for the last 5 years. International experts in the field will be contacted for comments and advice. Data extraction will include identification and methodological synthesis of each study; definition of recovery; information on recovery implementation; facilitators and barriers and study outcomes. A quality assessment will be conducted on each study. The data will be synthesised and a stepwise thematic analysis performed.

**Ethics and dissemination** Ethics approval is not required for this knowledge synthesis. Findings will be disseminated through knowledge translation activities including: (1) a 1-day symposium; (2) presentations in national and international conferences and to local stakeholders; (3) publications in peer-reviewed journals; (4) posts on the organisational websites.

### Strengths and limitations of this study

► Studies included in this knowledge synthesis will have been conducted in inpatient, outpatient and community-based mental health settings and will cover a broad range of research methodologies. The synthesis will reveal how recovery is understood; challenges involved in implementation and, overall, to what extent transformation to recovery oriented services and systems is occurring.

► The selection of recovery-oriented studies with an implementation focus is unique and will allow us to draw on a powerful conceptual model from implementation science that provides theory-informed elements to guide data analysis and synthesis as well as the reporting of results.

► Knowledge from the synthesis will be compiled into comprehensive and usable formats for organisational and governmental stakeholders, providing practical guidelines for recovery-based service reform and future evaluation.

► Limiting the search to published, peer-reviewed studies, while important for considerations of quality and methodological rigour, may overlook possible research on recovery-oriented services reported elsewhere.

## INTRODUCTION
### Rationale

Recovery is the focus of national mental health plans in G-8 countries[1–3] including Canada's first mental health strategy, Changing Directions, Changing Lives[4] and several provincial strategies.[5–8] The rationale for transformation to recovery-oriented services in mental health is compelling. While traditional mental health services have underlined professional control,[9–13] reinforcing patient dependency, self-stigma and hopelessness,[14–17] recovery approaches focus on individual empowerment, strong collaborative relationships between mental health service providers and service users and community integration.[18–24] In promoting a life beyond services, recovery also meets a key ethical obligation to honour the personhood and citizenship of people with mental illness.[25]

Recovery knowledge and evidence have burgeoned over the past two decades. Research exists on personal recovery,[26–32] recovery-oriented services[33–40] and provider competencies.[41–45] Conceptual frameworks and standardised measures have been produced.[35 46–49] Other studies have linked the recovery approach to recognised theories, such as empowerment theory,[50 51] the strengths model,[52] capabilities theory,[53–55] positive psychology,[56–58] person-centred practice[59 60] and coproduction.[61 62] Practice guidelines for recovery-oriented service provision are available,[63–69] as well. In terms of the empirical literature, studies on particular agencies and programmes have identified potential determinants of recovery orientation in services, for example, a flexible and innovative organisational culture, results-oriented leadership and larger budgets[70] were found to be associated with recovery-oriented services, as were provider socioprofessional characteristics such as greater age, higher educational levels and more professional experience.[71 72] Another recent study found that increasing the recovery orientation of teamwork on mental health teams[73] was associated with provider and consumer perceptions that services were recovery oriented.

While two recent systematic reviews have been conducted on the recovery-oriented practices of mental health service providers,[74 75] no known review has been published, to date, on the implementation of the recovery approaches into mental health services. Our review synthesizes research on the nature of recovery-oriented services, implementation challenges and overall system transformation. Work on the project was initiated in August 2016 and should come to completion in spring 2018.

This project is important and timely, as mental illness affects millions of people worldwide. According to recent WHO statistics,[76] 350 million people are impacted by depression; 60 million people by bipolar disorder and 21 million affected by schizophrenia. A recovery-oriented approach to healthcare is expected to reduce dependency and reduce the cost of mental healthcare, which in Canada has risen to more than $50 billion per year.[4 77 78] The project responds to a critical knowledge gap identified by knowledge users across Canada, who are responsible for implementing provincial level policy as well as shifting mental health organisations and services to a recovery orientation.

### Objectives
The overall goal of this review is to systematically search, assess and synthesise implementation studies on mental health recovery from the international mental health literature in order to inform and facilitate the transformation of Canadian mental health systems and adult services to a recovery orientation. The following six research questions, guided by the Consolidated Framework for Advancing Implementation Science (CFIR)[79] will be applied to each selected study: (1) How was recovery defined in this study?; (2) How was the recovery approach implemented in this study (Intervention)?; (3) What elements from the external environment (outer setting) or internal environment (inner setting), influenced implementation in this study?; (4) What were the characteristics of participants in this study? (characteristics of individuals); (5) What processes were involved in effecting the implementation? and (6) What was the extent and effectiveness of implementation in this study?

## METHODS
### Eligibility criteria
The design and methodology for the present review are reported following the Preferred Reporting Items for Systematic review and Meta-Analysis Protocols guidelines[80] (online supplementary appendix 1).

### Population
The review concerns studies of services for adults (≥18 years) with a primary diagnosis of schizophrenia, bipolar disorder or major depression, following the DSM-V (Diagnostic and Statistical Manual of Mental Disorders, 5th Edition) classification for mental disorders.

### Intervention
Studies will be included if they describe and evaluate the implementation of any intervention based on recovery principles that aims at transforming the orientation of mental health services or organisations to a recovery approach.

### Comparators
Studies will be eligible for inclusion whether or not they include comparison groups.

### Outcomes
Selected studies should report outcomes related to the transformation of a mental health service or organisation to a recovery orientation. Outcomes might include change in organisational culture; more integrated service networks and partnerships; increased knowledge, skills and/or attitudinal change among mental health providers; more use of evidence-based recovery-oriented best practices; greater consumer/provider collaboration, consumer self-management and evaluation.

### Study design
This will be a mixed studies review (MSR).[81 82] The MSR integrates qualitative, quantitative and mixed methods studies, providing a rich, detailed understanding of complex health interventions and programmes.[83] Studies representing a full range of methodologies will be included: systematic reviews and meta-analyses, randomised controlled trials (RCTs) and clinical trials, observational, mixed method and qualitative studies.

## Time period

We would expect to find very few pertinent studies prior to 1998, when recovery was first defined in an international policy document.[84] Thus, the precise time frame for the review is from 1998 to December 2016.

## Setting

Research settings may include inpatient, outpatient or community-based mental health services.

## Exclusion criteria

Non-research studies (eg, editorials, letters, conference abstracts), as well as unpublished (grey) literature, dissertations and book or book-length studies, will be excluded, as well as conceptual papers and review articles. Studies on services for addiction populations will also be excluded, as recovery is conceptualised differently in the addictions field. Language restrictions will not apply.

### Information sources

Our final search strategy will be developed in consultation with an experienced research librarian on the project and will combine a broad, systematic search of the literature. Electronic search will be conducted on the following databases: PubMed, Ovid Medline, Ovid Embase, Ovid PsycInfo, CINAHL (Current Index to Nursing and Allied Health Literature), the Cochrane Library and Scopus. We will supplement our results by conducting the following secondary searches: (1) Reference lists of all selected full text articles will be scanned for additional relevant studies; (2) Citation tracking will be performed on included articles; (3) Handsearches of tables of contents for the past 5 years will be conducted in the following key journals: *Psychiatric Rehabilitation Journal*, *Psychiatric Services* and *Community Mental Health Journal*. Additional journals will be added if warranted and (4) Known experts in the field will be contacted for comments and advice. We will also stay alert to serendipitous discovery that may increase results.

### Pilot search strategy

Using the Ovid Medline database, a research librarian and coinvestigator on the project conducted a pilot search (online supplementary appendix 2) which generated 5164 records. For this preliminary scoping phase, the search strategy was designed to focus on three main components: mental health, recovery and services. Medical Subject Heading and synonyms (keywords) were combined for each of the components. Terms related to recovery were chosen to reflect the consumer–survivor understanding of recovery.[48 85 86] While keywords will remain consistent throughout the searches, subject headings will be revised to reflect database-specific preferences. Search strategies will be further revised as new subject headings and keywords are revealed.

## STUDY RECORDS
### Data management

Electronic search results will be downloaded into EndNote reference manager software, duplicates will be removed where possible and the remaining references will be uploaded to the Distiller Systematic Review software for the screening and data extraction stages. Distiller software stores references, manages and monitors the screening and data extraction process with customised forms and automated flowcharts and provides an audit trail for the review.

### Screening and selection process

For the first selection, two team members working independently will read titles and abstracts of each paper identified in the electronic search and assess them for relevance based on the inclusion and exclusion criteria. Second, the team members will read the full text of each selected article in order to confirm its inclusion in the study. Disagreements related to the inclusion of any paper will be discussed and resolved, involving a third team member if necessary. To ensure high inter-rater reliability, training exercises will be conducted prior to initiating the screening process. Team members will meet on a weekly basis to follow-up on the screening process and discuss unanticipated problems.

### Data items and data extraction process

In order to minimise bias, two research team members will independently extract the data. Sample elements for data extraction appear below in table 1. The categories on the extraction grid include methodological elements based on the PICO mnemonic (PICO=population, intervention, comparison, outcome).[87] Also elements corresponding to the six research questions will be extracted and organised using the CFIR,[79] a multilevel five-dimension determinant framework[88] that constitutes a highly useful tool for identifying barriers and facilitators influencing implementation outcomes. Study limitations and gaps in knowledge will also be recorded.[89] The data extraction form will be pretested by the two reviewers and revised as needed. Distiller SR software will be used to manage the data extraction process.

### Quality assessment

Systematic reviews require that selected studies are assessed for quality.[90] We will use the Assessing Methodological Quality of Systematic Reviews tool (AMSTAR) protocol for the assessment of systematic reviews. The AMSTAR is an 11-item questionnaire that assesses study design, literature searched and scientific quality of reviews; a rating system is included. For primary research studies, quality assessment will be determined using criteria developed by Kmet *et al*.[91] This tool includes a 14-item checklist for quality criteria in quantitative studies and a 10-item checklist for qualitative studies. A rating system (yes-2; partial-1 and no-0) is provided, as well as a calculation for summary scores. While some controversy

**Table 1** Sample elements for data extraction

| Study ID/methods | Recovery definition and intervention | Characteristics of the setting | | Characteristics of individuals | Process | Implementation outcomes |
|---|---|---|---|---|---|---|
| | | Outer | Inner | | | |
| ID no; location<br>Objective<br>Study design<br>Participants<br>Interventions/comparisons<br>Data collection<br>Data analysis/triangulation<br>Quality appraisal | ▲ Recovery definition/ conceptualisation<br>▲ Intervention characteristics source adaptability trialability complexity design and packaging costs | ▲ Needs and resources<br>▲ External links<br>▲ Political pressures<br>▲ Policy and incentives | ▲ Structure<br>▲ Dialogue<br>▲ Culture<br>▲ Tension for change<br>▲ Values and norms<br>▲ Priorities<br>▲ Incentives and rewards | ▲ Knowledge and beliefs<br>▲ Identification with organisation<br>▲ Other personal attributes | ▲ Advance planning<br>▲ Buy-in Opinion leaders<br>▲ Implementation leaders Champions<br>▲ Execution<br>▲ Feedback | ▲ Extent of successful implementation of intervention<br>▲ Conclusions<br>▲ Study limitations/ gaps<br>▲ Contradictions/ further questions |

exists on whether qualitative research should be assessed using standard quality criteria,[92] we will include a quality appraisal for qualitative, as well as quantitative, studies in order to better assess the strengths and weaknesses of the evidence.[93] The Cochrane Collaboration tool will be used for assessing the risk of bias in RCTs.[94] The selected studies will be independently assessed for quality by two reviewers. Discrepancies will be solved in consultation with the principal investigator.

## DATA
### Data synthesis
No single unifying framework exists for synthesising quantitative and qualitative evidence for healthcare policy-makers and managers.[95] Our experience with recovery research suggests that much of the pertinent literature for review will be qualitative. Thus, our overall approach will be to convert all the evidence into qualitative form. The quantitative data will be transformed into qualitative form by extracting key concepts and findings within the elements geared to our research questions, as described above. Analytic procedures and synthesis will follow a three-stage process: (1) organisation of studies into logical categories according to their design, and methodology and coding using NVivo 11 software; (2) within-study analysis, according to the study questions; (3) cross-study synthesis of the data using an adaptation of the stepwise thematic analysis developed by Lucas et al,[93] according to the following procedures: (1) two reviewers will independently review data collated under each of the research questions; (2) codes produced by each researcher will be compared and a consolidated list of themes produced for each research question; (3) themes occurring under each question will be clustered around common dimensions; (4) results of the thematic analysis will be presented to the research team at a consensus meeting.

Specific measures will be taken to enhance the trustworthiness of the data. As suggested by Lucas et al, directly reported participant data (eg, verbatim quotations or scores on attitudinal scales) and author interpretations will be reported separately in order to retain the richness or 'thickness' of the data. Detailed descriptions, contextual material and the quality assessment of each paper will also help readers make judgements about the reliability and validity of the data. Summary tables will include counts of the papers contributing data on each theme.[92]

## DISSEMINATION
Knowledge translation will involve collaboration with our international consultants and knowledge users, who include decision makers and managers, service providers, people with lived experience and families. Four output documents will be developed, including: (1) a critical appraisal of findings from the synthesis on recovery implementation; (2) a compendium of case studies on successful recovery implementation

initiatives (sensitive to gender, race, culture); (3) a recovery implementation manual for decision makers and managers and (4) a toolkit of recovery-enhancing approaches, that is, strategies for individual behavioural change targeted at service providers and service users. Each document will be submitted to the entire team for revision and editing.

The results of the synthesis project will be widely disseminated. Knowledge translation activities will include: (1) creation of an Advisory Committee composed of the research team, knowledge users and international expert advisors. Quarterly telephone meetings with the Advisory to review emerging findings and provide feedback; (2) ongoing consultation/feedback between knowledge users and the research team during preparation of the four project outputs; (3) posting of information and updates on the websites of the organisations of knowledge users on the project, including those for people with lived experience and families; (4) organisation of a 1 day end-of-project symposium for dissemination of project outputs, including workshops for recording feedback and recommendations; (5) dissemination of project outputs through organisational websites and through national and international networks (free access); (6) submission of articles to peer-reviewed, open access journals; (7) presentations at national/international conferences.

## CONCLUSION

The recovery approach emerged through the lived experiences of people with enduring mental health problems as they used the formal mental healthcare system. Recovery-oriented services are viewed as a more person-centred and promising approach for treating mental illness. Until now, there has been little access to knowledge concerning how mainstream mental health services are being transformed to a recovery orientation and with what results. Our synthesis will establish the state of knowledge and evidence on implementing recovery and will make this knowledge available to a wide range of mental health stakeholders through dissemination activities and the publication of concrete recovery implementation tools. Results may also support the development of new recovery interventions, on which future outcome research should be considered.

**Author affiliations**
[1]Department of Psychiatry, McGill University, Montreal, Quebec, Canada
[2]School of Social Work, McGill University, Montreal, Quebec, Canada
[3]Douglas Mental Health University Institute, Research Centre, Montreal, Quebec, Canada
[4]Schulich Library of Physical Sciences, Life Sciences and Engineering, McGill University, Montreal, Quebec, Canada
[5]Mental Health Commission of Canada-Ottawa, Ottawa, Ontario, Canada
[6]Faculty of Medicine, University of Montreal, Montréal, Quebec, Canada
[7]School of Nursing, Faculty of Health Professions, Dalhousie University, Halifax, Nova Scotia, Canada

**Acknowledgements** This study was funded by the Canadian Institutes of Health Research Grant # KRS 144043

**Contributors** MP is the guarantor. MP, ES and JS drafted the manuscript. MP, ES, JS, CB, BV and JC contributed to the development of the selection criteria and data extraction criteria. AL developed the preliminary search strategy. HC, CB, BV and JC provided written feedback on the manuscript and AL, HC, CB, BV and JC approved the final manuscript.

**Funding** This study was funded by the Canadian Institute of Health Research, grant number KRS 144043.

**Competing interests** None declared.

**Provenance and peer review** Not commissioned; externally peer reviewed.

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
