## [Reviewer comments · BMJ Open]

ARTICLE DETAILS

TITLE (PROVISIONAL)	Protocol for a mixed studies systematic review on the implementation of the recovery approach in adult mental health services
AUTHORS	Piat, Myra; Sofouli, Eleni; Sabetti, Judith; Lambrou, Angella; Chodos, Howard; Briand, Catherine; Vachon, Brigitte; Curran, Janet

VERSION 1 - REVIEW

REVIEWER	Dr Clair Le Boutillier King's College London, UK
REVIEW RETURNED	08-May-2017

GENERAL COMMENTS	This is a very clear and well written paper. I have three suggestions for revisions/clarity: 1. The title indicates that the review will be specific to MHS in Canada. This is not explicit in the paper.2. The method states that studies will be included if they describe and evaluate any intervention based on recovery principles. Will the protocol use pre-defined terms to search for recovery principles (e.g. CHIME as one example of recovery processes)? A more specific inclusion/exclusion criteria would be useful.3. Similarly, data extraction/analysis will include Q1. meaning of recovery. Is this using a pre-defined framework? Recent studies suggest three types of recovery.
---

REVIEWER	Deede Gammon Center for Shared Decision-Making and Collaboration Care Research, Oslo University Hospital
REVIEW RETURNED	08-Jun-2017

GENERAL COMMENTS	The review described in this protocol is timely and ambitious and may be expected to provide important insights into recovery implementation given some revisions I have some suggestions for strengthening the protocol: 1) Abstract. Objectives (page 2/lines 15-21) should reflect the objectives outlined on pages 5 and 6. Write for example, "The objective of the study is to answer the research questions (RQ); a).b)...c)." The current text is about methods. In table 1 the 4 RQ is not included in the gg2) The different uses of 'recovery' together with – 'concepts', 'approaches', 'practices', 'orientations' - is somewhat confusing and needs more consistency. For example, what would it entail to
---

	implement a 'recovery concept', as opposed to an approach or practice? Isn't it the later that is of interest? 3) Many studies and interventions use the recovery concept, or claim to follow a recovery approach, while mainly adhering to biomedical principles. How will you determine what studies to include as "...any intervention based on recovery principles that aims at transforming the orientation of mental health services or organisations to a recovery approach" (p6, 25-30), and which to exclude? This should be elaborated on p.8 and 9 Screening and Selection process. 4) The research question 2 (How is the recovery approach implemented?) is very broad and doesn't seem to coincide with the theoretical frameworks in Table 1. What is the justification for selecting the 5 theoretical frameworks? What is it you are actually looking for? Those used are best suited to barriers and facilitators (CRIF) and implementation strategies (Powell). If implementation processes are the focus, other frameworks are likely more suitable. Per Nilsen's (2015) Making sense of implementation theories, models and frameworks may be of help here. 5) The conduct of a quality assessment of the reviewed studies, as described under methods, should probably be in response to a research question – What is the quality of the research?
--	---

VERSION 1 – AUTHOR RESPONSE

Reviewer: 1

Dr Clair Le Boutillier

King's College London, UK

Please state any competing interests or state 'None declared': None declared

Please leave your comments for the authors below

This is a very clear and well written paper. I have three suggestions for revisions/clarity:

1. The title indicates that the review will be specific to MHS in Canada. This is not explicit in the paper.

The title has been revised to correct the (false) impression that the review will be specific to Canada. Please refer to the editor's comments, point 5.

2. The method states that studies will be included if they describe and evaluate any intervention based on recovery principles. Will the protocol use pre-defined terms to search for recovery principles (e.g. CHIME as one example of recovery processes)? A more specific inclusion/exclusion criteria would be useful.

We are well aware of the CHIME framework, which provides the most reliable known set of constructs describing personal recovery. We are not aware of an existing framework, based on the same exhaustive review of the literature that would provide pre-determined elements for recovery-oriented services. Rather than extrapolate a conceptual model from one of the more comprehensive sets of recovery guidelines (e.g. the 2013 Australian framework), and restrict ourselves to pre-determined elements, we will include any implementation studies where the interventions were defined by the authors as "recovery-oriented", or "based on recovery principles". Part of our task will be to analyze how recovery is defined, and operationalized in these selected studies.

3. Similarly, data extraction/analysis will include Q1. meaning of recovery. Is this using a pre-defined framework? Recent studies suggest three types of recovery.

Again, while we are aware of the 3 types of recovery identified in the systematic review by Le Boutillier et al, 2015 (clinical recovery, personal recovery, service-defined recovery), we decided to compile, and assess, the various meanings of recovery as they occur within this body of research, from the perspectives of the authors. Our expectation is that this approach might shed additional light on the meaning of “service defined recovery”, for instance, and will indicate how this concept has been operationalized in terms of specific recovery interventions. A comparison of our study with LeBoutillier, 2015 should make for an interesting discussion. Thank you for bringing this article to mind, and for your thoughtful review!

Reviewer: 2

Deede Gammon

Center for Shared Decision-Making and Collaboration Care Research, Oslo University Hospital

Please state any competing interests or state 'None declared': None declared

Please leave your comments for the authors below

The review described in this protocol is timely and ambitious and may be expected to provide important insights into recovery implementation given some revisions

I have some suggestions for strengthening the protocol:

1) Abstract. Objectives (page 2/lines 15-21) should reflect the objectives outlined on pages 5 and 6. Write for example, “The objective of the study is to answer the research questions (RQ); a)..b)...c)..” The current text is about methods. In table 1 the 4 RQ is not included in the gg

This point is no longer relevant, as the reviewers were working with the wrong version of the Abstract. Please refer to the Editor`s comments, point 2.

2) The different uses of ‘recovery’ together with – ‘concepts’, ‘approaches’, ‘practices’, ‘orientations’ - is somewhat confusing and needs more consistency. For example, what would it entail to implement a ‘recovery concept’, as opposed to an approach or practice? Isn’t it the later that is of interest?

We agree with the reviewer’s point that the use of terms should be consolidated. We eliminated the term “concept”, which is less apt in the context. We are not including studies on recovery oriented “practices”, as two systematic reviews of recovery-oriented practices already exist (ref: page 4, lines 90-92. We have revised the text to include use of the terms “approach” or “orientation” for better consistency and clarity.

3) Many studies and interventions use the recovery concept, or claim to follow a recovery approach, while mainly adhering to biomedical principles. How will you determine what studies to include as “...any intervention based on recovery principles that aims at transforming the orientation of mental health services or organisations to a recovery approach” (p6, 25-30), and which to exclude? This should be elaborated on p.8 and 9 Screening and Selection process.

You are quite correct in articulating this concern. We were careful to develop the pilot search strategy

using recovery-related terms that reflected recovery as understood from the consumer-survivor perspective. We have added a sentence underlining this point (Pilot Search Strategy). However, articles did emerge from the preliminary search that used “recovery” to describe the traditional, medical-model perspective. Yet, in all such cases the outcome of interest was personal recovery, not the recovery-orientation of services, and these studies were excluded on that basis.

4) The research question 2 (How is the recovery approach implemented?) is very broad and doesn't seem to coincide with the theoretical frameworks in Table 1. What is the justification for selecting the 5 theoretical frameworks? What is it you are actually looking for? Those used are best suited to barriers and facilitators (CRIF) and implementation strategies (Powell). If implementation processes are the focus, other frameworks are likely more suitable. Per Nilsen's (2015) Making sense of implementation theories, models and frameworks may be of help here.

We agree with this point, and thank the reviewer very much for bringing the Nilsen study to our attention. We have decided on the basis of this article to use the Damschroder framework exclusively, as the best fit with the aims of the review, and a highly useful tool for understanding, and explaining, factors influencing implementation outcomes. Accordingly, we have revised both the research questions, and the preliminary data extraction grid, integrating the dimensions of the Damschroder framework, with slightly adapted sub-dimensions, which appear as elements under “recovery intervention, “characteristics of the setting, and “barriers/facilitators. Depending on the nature of the studies that emerge in the full review, we may add a second framework, e.g. Powell.

5) The conduct of a quality assessment of the reviewed studies, as described under methods, should probably be in response to a research question – What is the quality of the research?

Since quality assessment is required in systematic reviews, we have declined to develop a specific research question, and reference Aromataris, 2015 in this connection (section quality assessment, page 9, line 206. Many thanks for your insightful comments that helped us improve the manuscript.

VERSION 2 – REVIEW

REVIEWER	Deede Gammon Center for Shared Decision-Making and Collaborative Care Research, Oslo University Hospital, Norway
REVIEW RETURNED	10-Jul-2017
GENERAL COMMENTS	Objectives need to be included in the abstract. On line 119, insert 'was' initiated.. On line 235, insert 'use' the Assessing...

VERSION 2 – AUTHOR RESPONSE

Dear Dr Deede Gammon,
We have made all changes requested to our manuscript. This includes:

Objectives - now in Abstract
Line 119 - the word "was" is inserted
Line 235 - the word "use" is inserted